# Large near-term projected snowpack loss over the western United States

John C. Fyfe[1], Chris Derksen[2], Lawrence Mudryk[2], Gregory M. Flato[1], Benjamin D. Santer[3], Neil C. Swart[1], Noah P. Molotch[4,5], Xuebin Zhang[2], Hui Wan[2], Vivek K. Arora[1], John Scinocca[1] & Yanjun Jiao[1]

Peak runoff in streams and rivers of the western United States is strongly influenced by melting of accumulated mountain snowpack. A significant decline in this resource has a direct connection to streamflow, with substantial economic and societal impacts. Observations and reanalyses indicate that between the 1980s and 2000s, there was a 10–20% loss in the annual maximum amount of water contained in the region's snowpack. Here we show that this loss is consistent with results from a large ensemble of climate simulations forced with natural and anthropogenic changes, but is inconsistent with simulations forced by natural changes alone. A further loss of up to 60% is projected within the next 30 years. Uncertainties in loss estimates depend on the size and the rate of response to continued anthropogenic forcing and the magnitude and phasing of internal decadal variability. The projected losses have serious implications for the hydropower, municipal and agricultural sectors in the region.

[1] Canadian Centre for Climate Modelling and Analysis, Environment and Climate Change Canada, Victoria, British Columbia, Canada V8W 2Y2. [2] Climate Research Division, Environment and Climate Change Canada, Toronto, Ontario, Canada M3H 5T4. [3] Program for Climate Model Diagnosis and Intercomparison, Lawrence Livermore National Laboratory, Livermore, California 94550, USA. [4] Institute of Arctic and Alpine Research, University of Colorado, Boulder, Colorado 80309-0450, USA. [5] Jet Propulsion Laboratory, California Institute of Technology, Pasadena, California 91011, USA. Correspondence and requests for materials should be addressed to J.C.F. (email: John.Fyfe@canada.ca).

It is well established that the North American continent is warming[1], partly due to increasing emissions of well-mixed greenhouse gases[2]. In the winter season, this warming contributed to snowpack loss over the western United States[3–6]. It is also well known that the region's climate is substantially influenced by decadal variability originating in the adjacent Pacific Ocean[7]. In this study we employ observations, land surface reanalyses and climate model simulations to characterize the combined influences of decadal variability and external forcing on recently observed and near-term projected changes in snowpack over the western United States.

Recognition of the pronounced influence of decadal variability on regional trends in climate is motivating efforts to generate large initial condition ensembles of global climate model simulations. These ensembles provide estimates of the relative contributions of internal variability and external forcing to regional-scale climate changes[8–11]. An initial condition ensemble consists of many individual simulations performed with a given coupled climate model; each simulation uses the same external forcing, but is initiated from slightly different conditions of the atmosphere and/or ocean state. Each ensemble member has a different realization of internal variability superimposed on the underlying externally forced response.

Because of their high computational cost, large (>10-member) ensembles are rare. It is also rare for groups to perform multiple large ensembles, one consisting of simulations with anthropogenic forcing only, and one with simulations incorporating solar and volcanic forcing alone. Here, we generate and analyse 50-member anthropogenic and naturally forced ensembles. We also rely on a large ensemble of higher-resolution regional climate model simulations driven with the output from the global climate model.

Although several studies have found an anthropogenic contribution to snowpack loss over the western United States, the combined influences of decadal variability and external forcing remain poorly quantified in observations and near-term projections. In this study, we show that losses in regional snowpack over the past few decades are consistent with natural and anthropogenic changes, but are inconsistent with natural changes alone. We predict an additional loss of snowpack water storage of up to 60% within the next three decades due to combined influences from anthropogenic forcing and internal decadal variability.

## Results

**Observed snowpack loss.** Observations of snow water equivalent (SWE) were acquired from the United States Natural Resource Conservation Service Snow Telemetry (SnoTel) network of automated snow pillow measurements across alpine sites. These measurements were taken from a quality-controlled data set discussed in ref. 12. We restrict our analysis to the post-1981 period since the number of observations before this time is too low to allow calculation of reliable regional averages. Monthly-mean (January-May) SnoTel observations are continuously available from 1982 to 2016 at 354 stations with elevations greater than 1,500 m. A threshold of 1,500 m provides good regional coverage; use of continuous observations avoids introducing temporal changes in spatial coverage. Figure 1 shows that 307 of the 354 stations (or about 87% of all stations) show a negative trend in annual maximum SWE ($SWE_{max}$). The maximum loss typically occurs in April. Here and subsequently, $SWE_{max}$ is computed over 1982–2010 to facilitate comparison with reanalysis-based estimates for the maximum period of overlap between the reanalyses and the SnoTel data (see below).

We first compute the climatology and trend in $SWE_{max}$. Results are area-averaged over the 354 selected SnoTel stations. Figure 2 shows that the climatology and trend are 4.6 cm and $-0.33$ cm per decade, respectively. The trend has a $P$ value less than 0.15. This decrease in $SWE_{max}$ represents about a 9.5% loss; this is expressed as the per cent difference between the decadal averages of $SWE_{max}$ in 1982–1992 and 2000–2010 (the first and last decades of the period of maximum overlap). This multi-decade decline in regional snowpack is consistent with results from refs 13–15, despite the fact that these studies used different time periods, data sets and metrics. (The sensitivity to the metric used, for example, 1 April SWE or $SWE_{max}$, has been discussed in ref. 16.) Finally, we note that the accumulated SWE over the winter months is primarily controlled by snowfall. Significant losses in SWE are observed throughout the snowfall season (Supplementary Fig. 1) and are not restricted to April, the month typically associated with $SWE_{max}$.

We also consider a group of four gridded SWE data sets obtained from reanalysis products (hereafter referred to as REANAL4). These four data sets have been assessed in ref. 17 over the common period of availability from 1982 to 2010. Whereas the SnoTel measurements provide a purely observational perspective, the REANAL4 group has been produced using snow schemes of varying complexity embedded in land surface models forced by meteorological reanalyses. The following data sets were used: (1) the Global Land Data Assimilation System Version 2 (ref. 18); (2) the European Centre for Medium-Range Forecasts Interim Land Reanalysis[19]; (3) the Modern Era Retrospective Analysis for Research and Applications[20]; and (4) the Crocus snow scheme driven by ERA-Interim meteorology[21]. Differences between these data sets have been discussed at length in ref. 17. The SWE fields were re-gridded to a 1° longitude by 1° latitude grid; monthly mean SWE results were averaged across the four reanalyses.

Area averaging $SWE_{max}$ at elevations greater than 1,500 m (north of 30°N and south of 50°N) yields a reanalysis-mean climatology and trend of 3.9 cm and $-0.56$ cm per decade, respectively. The trend has a $P$ value less than 0.01. This decrease in $SWE_{max}$ represents an $\sim 21.8\%$ loss between 1982–1992 and 2000–2010. The correlation between the $SWE_{max}$ time series from REANAL4 and SnoTel is 0.73. The range of climatology and trend estimates from the four reanalysis products encompasses the corresponding SnoTel estimates (Fig. 2). While this is encouraging, there are significant scale challenges in relating local SWE measurements to coarsely gridded snow reanalyses, and direct comparisons between the two must be approached cautiously[22]. The green bars in Fig. 2 show the uncertainties associated with the SnoTel values. Here, uncertainties are computed as $\pm 1$ s.d. across 10,000 random samples of the SnoTel stations. In each case, a station is randomly selected and the average climatology and trend is computed across a random selection of no more than 10 of its neighbouring stations within 1° longitude and 1° latitude (the spatial resolution of the reanalyses). In short, the SnoTel and REANAL4 results are consistent within the uncertainties in each data source; this holds for both climatologies and trends.

**Historical simulated snowpack loss.** Our ensemble of model simulations was generated with the Canadian Earth System Model version 2 (CanESM2; see Methods and ref. 23). Simulations include historical forcings from 1950 to 2005 and RCP8.5 forcing extensions after 2005. The RCP8.5 pathway is a high-emission scenario leading to an 8.5 W m$^{-2}$ increase in radiative forcing by 2,100. Over the early part of the twenty-first century, forcing differences between RCP8.5 and other commonly

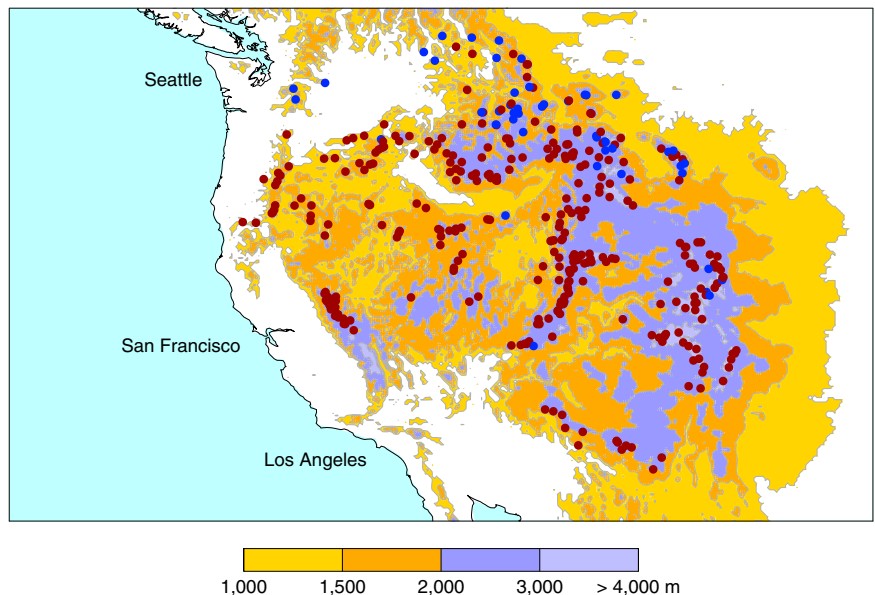

**Figure 1 | Topography and measurement network.** The circles are the snow telemetry (SnoTel) network of stations utilized in this study. The red circles denote stations with negative trends in annual maximum snow water equivalent ($SWE_{max}$). The blue circles indicate stations with positive trends. Linear trends are computed from 1982 to 2010 to facilitate comparison with reanalysis-based estimates (for the period of maximum overlap between SnoTel and the four reanalysis products). All the stations plotted here are at elevations greater than 1,500 m.

analysed RCPs are relatively small[24]. Two 50-member ensembles were run with CanESM2, one with natural and anthropogenic forcings (ALL) and one with only natural forcings (NAT). Anthropogenic forcings include changes in well-mixed greenhouse gases, anthropogenic aerosols, tropospheric and stratospheric ozone, and land use. Natural forcings consist of changes in solar irradiance and volcanic aerosol loadings.

For the ALL ensemble, area-averaging the simulated $SWE_{max}$ at elevations greater than 1,500 m (north of 30°N and south of 50°N) yields an ensemble-mean climatology and trend of 6.1 cm and −0.57 cm per decade. The trend has a *P* value less than 0.001. The simulated trend is in good agreement with the average of the four reanalyses; the simulated climatology is at the high end of the reanalysis estimates (Fig. 2). The latter suggests greater climatological snowfall in the model than in reality, although this is difficult to assess given the absence of a reliable long-term observational record of winter precipitation. The simulated pattern of climatological $SWE_{max}$ is in reasonable agreement with the comparable field in reanalyses (see Fig. 3), albeit with spatial detail given the coarser resolution of the global coupled model (which is nominally 2.8° longitude by 2.8° latitude).

Figure 4a shows 5-year mean anomalies of area-averaged $SWE_{max}$ at elevations greater than 1,500 m. Results are from SnoTel (green), REANAL4 (pink) and the ALL simulations (black). We also show results obtained by dynamical downscaling of 35 members of the CanESM2 ALL ensemble and RCP8.5 continuation (dashed curve). Downscaling relied on the Canadian Regional Climate Model version 4 (CanRCM4), which has a nominal resolution of 50 km (see Methods and ref. 25). Downscaling was performed over the North American domain defined in the Coordinated Regional Climate Downscaling Experiment[26]. The similarity between the CanESM2 and CanRCM4 ensemble-mean time series ($r = 0.99$) is evidence that the temporal variability of area-average $SWE_{max}$ from CanESM2 is relatively insensitive to an increase in spatial resolution.

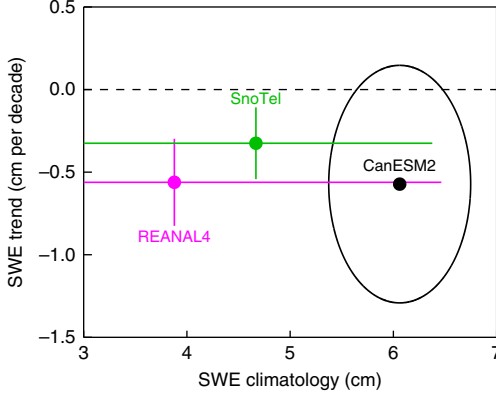

**Figure 2 | Climatology and trend in area-average annual maximum snow water equivalent.** The green bars show snow telemetry (SnoTel) ranges (±1 s.d.) computed from random samples of stations from the SnoTel network (see main text). The pink bars show ranges (±1 s.d.) from four land surface reanalyses (REANAL4). The black ellipse encompasses 95% of the climatology and trend values across 50 simulations of a global climate model. The circles represent the average climatology and trend for SnoTel, REANAL4 and the CanESM2 ALL simulations. All values in this figure are area averages of annual maximum snow water equivalent ($SWE_{max}$) at elevations greater than 1,500 m and north of 30°N and south of 50°N. The period considered is the maximally overlapping period from 1982 to 2010.

Because there is only one realization of internal variability in the real world, we do not expect the CanESM2 ensemble-mean $SWE_{max}$ (which is averaged over many different model realizations of internal variability) to closely follow the time evolution of $SWE_{max}$ in the SnoTel data and REANAL4. The key point here is that the multi-decadal changes in $SWE_{max}$ are reasonably similar in SnoTel, REANAL4, and the CanESM2 and CanRCM4 ensembles under ALL forcing. Under NAT forcing, however, the CanESM2 ensemble mean

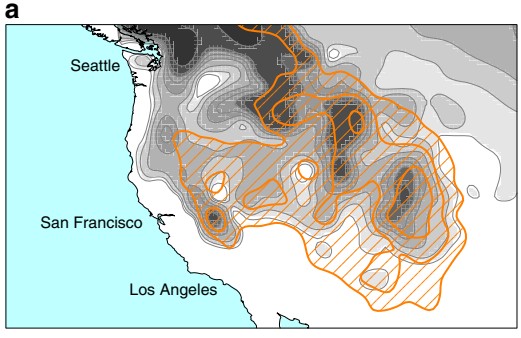

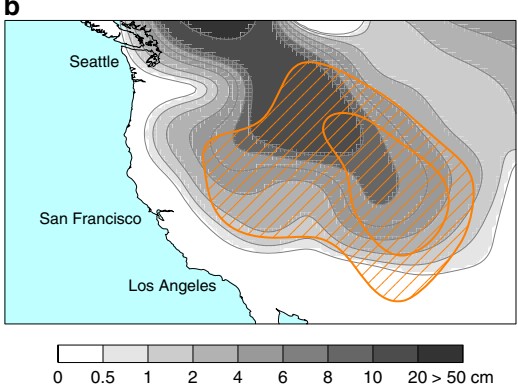

0  0.5  1  2  4  6  8  10  20 > 50 cm

**Figure 3 | Climatology of annual maximum snow water equivalent.**
(**a**) Average of four land surface reanalyses. (**b**) Average of 50 global climate model simulations from the ALL ensemble. The period considered is that of maximal overlap between the reanalysis products (1982–2010). The orange contours are topographic height in kilometers. The outer contour is 1,500 m and the inner contours are in increments of 500 m.

does not replicate the observed decline in $SWE_{max}$ since the 1980s.

**Detection and attribution analysis.** We use a standard optimal fingerprinting method to address the causes of these $SWE_{max}$ changes. This involves regressing the observations onto the simulated ALL and NAT responses. The resulting values of the scaling factor $\beta$ provide information on the extent to which the model $SWE_{max}$ responses must be scaled to best reproduce the observed $SWE_{max}$ changes. The method also yields the 90% confidence intervals on $\beta$ estimates (see Methods).

Figure 4b shows that with combined anthropogenic and natural forcings, CanESM2 reproduces the reanalysis and SnoTel $SWE_{max}$ changes, albeit with smaller magnitude—that is, the scaling factors are significantly larger than 0 and less than (but still consistent with) 1. With natural forcings alone, however, the model does not reproduce the reanalysis and SnoTel changes. A similar result was obtained in ref. 13 using snow course data, observationally based precipitation, two earlier-generation climate models, two statistical downscaling approaches, and a late-twentieth-century analysis period (1950–1999).

Since we do not have a CanESM2 ensemble with anthropogenic forcing only, we compute the anthropogenic scaling factor by linear regression of the reanalyses or observations onto the difference between the ALL and NAT responses (denoted by ANT). We assume additivity of the ANT and NAT $SWE_{max}$ responses. Figure 4b shows that the estimated ANT $SWE_{max}$ changes: (1) are consistent with the reanalyses; (2) are detectable in the SnoTel observations but with significantly smaller

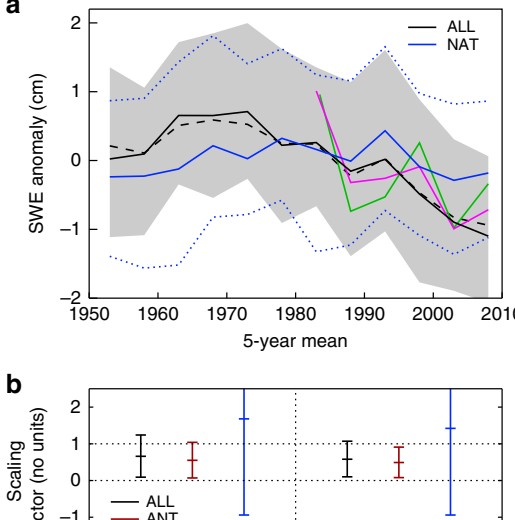

**Figure 4 | Anomaly of annual maximum snow water equivalent and scaling factor $\beta$.** (**a**) Anomaly in non-overlapping 5-year averages of annual maximum snow water equivalent ($SWE_{max}$). Solid black is ALL ensemble-mean and grey is 10–90% range (based on 50 ALL realizations). Solid blue line is NAT ensemble mean and dashed blue lines indicate the 10 and 90% values. The ALL and NAT curves are from CanESM2. The dashed black curve is from the 35-member set of CanRCM4 simulations with ALL forcing. Pink denotes the average of four reanalyses and green is the SnoTel observations. Anomalies are defined in Methods. (**b**) Scaling factor and 5–95% uncertainty range estimated from application of an optimal fingerprint method to SnoTel observations, reanalyses and CanESM2 simulation output (see Methods). Scaling factors greater than 0 and consistent with 1 indicate that a model-predicted $SWE_{max}$ signal has been detected in observations and attributed to the imposed forcing changes in the ALL or NAT simulations.

magnitude; and (3) yield very similar $\beta$ values to the ALL case. Regarding (2), when average $SWE_{max}$ is computed over the larger region encompassing elevations greater than 1 km (in order to reduce the noise contribution inherent in point measurements) anthropogenic forcing more successfully reproduces the magnitude of the SnoTel changes (Supplementary Fig. 2). While the results from this analysis provide clear evidence of an anthropogenic influence on snowpack water storage, it is difficult to more reliably quantify the magnitude of this influence given the relatively short observational $SWE_{max}$ record, the effects of underlying variability and the uncertainties inherent in our indirect estimate of the ANT $SWE_{max}$ response.

**Near-term projected snowpack loss.** Figure 5a shows 5-year mean $SWE_{max}$ from the ALL ensemble, averaged at elevations greater than 1,500 m. The first 5-year period is centred on 1988 and the last 5-year period is centred on 2038. The ALL ensemble uses the RCP8.5 forcing extension after 2005. The ensemble-mean response (black curve) shows a small increase in the rate of $SWE_{max}$ decline from the first half of the analysis period shown in Fig. 5a to the second half of the analysis period. More specifically, the externally forced trend in $SWE_{max}$ is about $-0.50$ cm per decade from 1988 to 2013 and about $-0.62$ cm per decade from 2013 to 2038. This increase in the rate of $SWE_{max}$ decline is significant at $P < 0.05$, where significance is assessed using a standard difference of means test on the

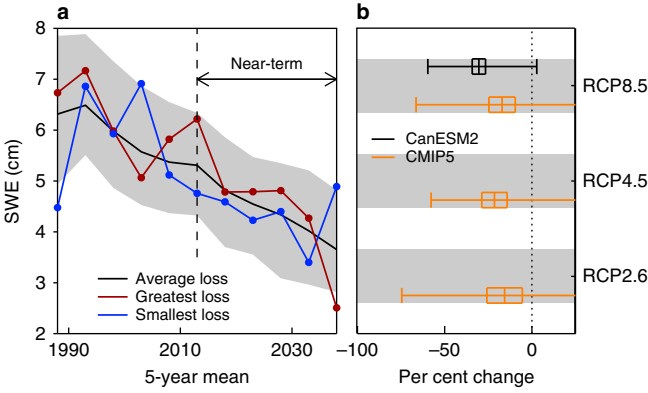

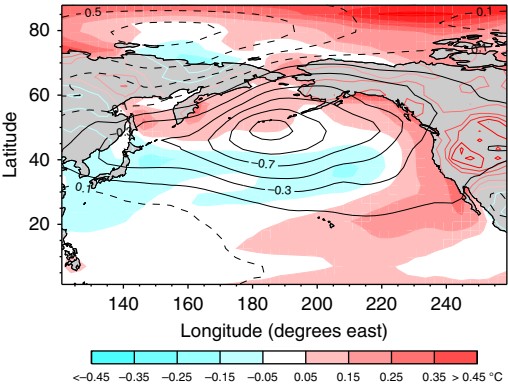

**Figure 5 | Annual maximum snow water equivalent and per cent change.**
(**a**) The black curve is for the CanESM2 ALL ensemble mean and shading is the 5–95% range. The ALL ensemble uses RCP8.5 forcing extensions after 2005. The red curve is the simulation with the greatest loss in annual maximum snow water equivalent ($SWE_{max}$) between 2013 and 2038. The blue curve is the simulation with the largest gain over the same 2013 to 2038 period. All values are 5-year averages plotted on the central year. (**b**) Per cent change in $SWE_{max}$ between the periods centred on 2013 and 2038 under different radiative forcing scenarios. Results in orange are from the CMIP5 multi-model ensemble (Supplementary Table 1). CMIP5 results are based on the analysis of one ensemble member (r1i1p1) from each CMIP5 model for which there exists a historical simulation with ALL forcing and a corresponding RCP2.6, RCP4.5 and/or RCP8.5 extension. The box-and-whiskers plots show ensemble-mean $SWE_{max}$ values, the 95% uncertainty ranges on the ensemble-mean values, and the minimum-to-maximum ranges. All values in this figure are for area averages of $SWE_{max}$ at elevations greater than 1,500 m and north of 30°N and south of 50°N.

**Figure 6 | Climate anomaly associated with large projected decease in regional snowpack.** Difference in winter (January, February and March) mean sea level pressure (contours in hPa) and surface air temperature anomalies (shading) between the five CanESM2 ensemble members with the lowest winter $SWE_{max}$ averaged over 2030–2040 and the five ensemble members with the highest winter $SWE_{max}$ averaged over the same period.

50 simulated trends in the earlier period and the 50 simulated trends in the later period. As noted in ref. 15, distinguishing rates of change in externally forced response over small regions and short timescales, such as these, requires large ensembles of simulations (as shown here).

Our focus is on quantifying the combined contribution of internal variability and external forcing to near-term projected $SWE_{max}$. The largest near-term loss in accumulated snowpack over the western United States is ∼60%; this was calculated as the per cent change between averages over the 5-year periods centred on 2013 and 2038. The largest near-term gain is about 3%. The substantial difference between these two percentage changes—obtained with the same physical climate model, driven by the same external forcings—underscores the potentially large contribution of decadal variability to regional-scale changes in $SWE_{max}$, particularly on shorter (ca. 30-year) timescales. The range in projected snowpack reductions reflects the interplay between temperature and precipitation trends on the end of season $SWE$[15,27,28]. Averaging over realizations reduces this variability, and provides a better estimate of the underlying response to external forcing, yielding about a 30% loss of regional snowpack in the ALL ensemble mean.

We note that CanESM2 generally reproduces the large-scale wintertime patterns of decadal variability observed in Pacific sea surface temperature, precipitation, sea level pressure and North American surface temperature[11]. This enhances our confidence in the credibility of the simulated decadal variations in snowpack loss. In comparison with other climate models participating in phase 5 of the Coupled Model Intercomparison Project phase 5 (CMIP5; Supplementary Table 1), the magnitude of unforced decadal variability in $SWE_{max}$ in CanESM2 is in the

bottom half of the multi-model ensemble (Supplementary Fig. 3). This suggests that the CanESM2 estimate of the influence of decadal variability on regional snowpack is conservative. Alternately, smaller decadal variability also has implications for the detection results in Fig. 4, and may yield more liberal estimates of anthropogenic signal detectability.

We also compare the projections of regional $SWE_{max}$ changes in CMIP5 models and CanESM2 (Fig. 4b). Results are for changes in ensemble-mean simulated $SWE_{max}$ between the 5-year periods centred on 2013 and 2038; the minimum-to-maximum ranges are also indicated. The CMIP5 set consists of one ensemble member (r1i1p1) from each CMIP5 model with an ALL forcing historical simulation and corresponding RCP2.6, RCP4.5 and RCP8.5 extension (Supplementary Table 1). The range associated with the CanESM2 initial condition ensemble (for ALL forcing and the RCP8.5 extension) solely reflects the influence of decadal variability. The ranges associated with the CMIP5 multi-model ensemble are indicative of both decadal variability and model uncertainty.

Although the 'spread' in these ensembles arises from different reasons, the message from this comparison is that the potential for large snowpack loss in the CanESM2 initial-condition ensemble is also evident in the CMIP5 multi-model ensemble, and is manifest across all three emissions scenarios.

**Large-scale patterns associated with extreme snowpack loss.** Figure 6 shows the sea level pressure and surface air temperature patterns associated with a large negative contribution to regional snowpack from decadal variability in the period from 2030 to 2040 as obtained from the CanESM2 ensemble. These patterns are indicative of a stronger than normal Aleutian Low. The Aleutian Low is a semi-permanent low pressure centre located near the Aleutian Islands during the winter. It is one of the main centres of action in the atmospheric circulation of the Northern Hemisphere. The circulation anomalies associated with the stronger than normal Aleutian Low are responsible for transporting anomalously warm and moist air over the western United States. The impact of warm air on regional snowpack is obvious (especially where temperatures are above or about the freezing point). The impact of increased precipitation on snowpack is generally more complex[29], but our model shows that the anomalous precipitation falls as rain rather than snow, therefore yielding a reduction in snowpack.

## Discussion

Our analysis makes innovative use of a large ensemble of simulations. This ensemble was generated with the same climate model and external forcings, but with each ensemble member commencing from slightly different initial conditions. The ensemble provides estimates of the relative contributions of internal variability and external forcing to near-term loss of regional snowpack over the western United States. Over this region, and over a relatively short (ca. 30-year) time horizon, decadal variability may offset some of the anthropogenically forced loss in snowpack. Of greater concern, however, is that decadal variability has the potential to substantially enhance the near-term decline in snowpack expected in response to anthropogenic forcing. These reductions in snowpack water storage have broad implications for future forest productivity and carbon storage[30], forest vulnerability to fire[31], as well as streamflow and water supply[32,33]. Such sensitivities should be carefully considered in mitigating climate risks, particularly in the context of water resource and land management in the western United States.

## Methods

**Global climate model simulations.** The simulations analysed here were performed with the CanESM2 (ref. 23), a climate model with interactive atmosphere, ocean, sea ice, land and carbon cycle components, run at T63 resolution. Two large initial-condition ensembles, each consisting of 50 simulations, were randomly initiated from the conditions on 1 January 1950. The random perturbation to the initial atmospheric state is introduced via a parameterization of one aspect of model cloud properties. This parameterization employs a random number generator with a pre-set seed; the 50 individual simulations in each ensemble were based on different seeds. In this way, different climate change realizations were produced without any change to the model dynamics, physics or structure. The NAT simulations used observed estimates of historical changes in solar irradiance and volcanic aerosol loadings. The ALL simulations incorporate (in addition to solar and volcanic forcing) estimated historical changes in greenhouse gases, aerosols, ozone and land use. Both ALL and NAT simulations end in December 2004. The RCP8.5 scenario was used to extend the ALL simulations from January 2005 onwards.

**Regional climate model simulations.** These simulations were performed with the CanRCM4; (this has a nominal resolution of 50 km; ref. 25). An ensemble of 35 CanRCM4 simulations was driven (at the horizontal and lower boundaries of the regional model's domain) by daily output from the large initial-condition ensemble of global climate model simulations. Observed natural and anthropogenic forcings are identical to those employed in the global coupled climate model simulations. Downscaling was performed over the North American domain defined in the Coordinated Regional Climate Downscaling Experiment[26].

**Detection and attribution analysis.** We use total least-squares linear regression to conduct a detection and attribution analysis[34]. The regression is expressed as
$$y = y^* + \varepsilon_0, \quad x_i = x_i^* + \varepsilon_i \quad \text{and} \quad y^* = \sum_{i=1}^{m} \beta_i x_i^*,$$
where $y$ is the observed time series, and each $x_i$ represents the model simulated response (or 'signal') to one of the $m$ forcings. The quantity $\varepsilon_0$ represents internal variability in the observations. The quantity $\varepsilon_i$ is noise in the signal $x_i$ arising from model internal variability that is not averaged out with the finite number of model simulations that are available.

The observational data include REANAL4 for 1982–2010 and SnoTel for 1982–2014. The anomalies relative to 1951–1980 climatology are used as $y$. For the REANAL4 and SnoTel series, the anomalies are obtained by removing the 1951–1980 climatology of the Global Land Data Assimilation System data set and the difference between REANAL4 or SnoTel over the common period 1982–2010, respectively, assuming time-invariant systematic biases among the data sets.

The ensemble averages of the 50 ALL simulations and the 50 NAT simulations provide estimates of the $SWE_{max}$ responses to combined anthropogenic and natural forcing and natural forcing only (respectively). Ensemble averages are expressed as anomalies relative to the model climatology for 1951–1980. These anomalies are our estimates of the signals $x_i$ in response to ALL and NAT forcings.

We assume that the model-simulated variability on the space and time scales used in the detection analysis is representative of the internal variability in the observations, $\varepsilon_0$. The validity of this assumption is examined through a residual consistency test[35]. Model internal variability estimates are from multiple sources: (1) the inter-ensemble variability of CanESM2 simulations under ALL, NAT and AER forcing; and (2) the pre-industrial control simulations listed in Supplementary Table 1. The AER ensemble consists of 50 members with anthropogenic aerosol forcing only. The components considered include sulfate aerosols, organic aerosols and black carbon. Half of the noise data are used for the estimation of internal variability, while the

remaining half is used for testing the estimation of the scaling factors. The estimation of the covariance matrix for $\varepsilon_0$ is based on a regularized covariance matrix[36].

The analyses are conducted on non-overlapping 5-year average time series of annual $SWE_{max}$ anomalies. We use 5-year mean time series to remove the influence of high-frequency natural variability, particularly variability related to the El Niño-Southern Oscillation. For REANAL4 and SnoTel, the first average is computed over the 4-year period 1982–1985.

**Data availability.** The CanESM2, reanalysis and SnoTel data associated with this paper are available on request from J.C.F., L.M. and N.P.M., respectively. The CMIP5 data are available at http://cmip-pcmdi.llnl.gov/cmip5/availability.html.

**Code availability.** The code associated with this paper is available on request from J.C.F.

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

## Acknowledgements

We thank Alex Cannon for helpful comments on the manuscript. We acknowledge the World Climate Research Programme's Working Group on Coupled Modelling, which is responsible for CMIP, and we thank the climate modeling groups for producing and making available their model output. For CMIP, the US Department of Energy's Program for Climate Model Diagnosis and Intercomparison provides coordinating support and led development of software infrastructure in partnership with the Global Organization for Earth System Science Portals. We acknowledge Environment and Climate Change Canada's Canadian Centre for Climate Modelling and Analysis for executing and making available the CanESM2 large ensemble simulations, and the Canadian Sea Ice and Snow Evolution (CanSISE) Network for proposing the simulations.

## Author contributions

J.C.F. conceived and designed the study, undertook the analyses, produced the figures and wrote the paper. C.D. and L.M. provided the reanalysis output, N.C.S. provided the CMIP5 multi-model output. N.P.M. provided the SnoTel data, and J.S. and Y.J. provided the regional climate model simulations. X.Z. and H.W. performed the attribution and detection analysis. All co-authors helped edit the paper.

## Additional information

**Competing interests:** The authors declare no competing financial interests.

**Publisher's note**: 

