## [Peer Review File · Nature Communications]

Reviewers' comments:

Reviewer #1 (Remarks to the Author):

The paper addresses an interesting and timely topic and the paper is cleanly written and compact. I only have two comments for the authors to consider (one is minor and one is major).

1. The authors present correlation statistics to indicate agreement between measured and modeled time series. Additional statistics that illustrate biases also are needed, or the authors can provide a justification for primarily focusing on variability rather than differences in magnitudes between measured and modeled time series.

2. The authors analyze annual maximum SWE. It would strengthen the paper if the authors also analyzed total winter snowfall. It could be that annual maximum SWE has decreased as shown in the study, but it also could be that more moderate SWE values have increased or become more prevalent for other moths (e.g. the temporal distribution of snowfall and SWE during winter has become flatter). An examination of total winter snowfall will confirm or refute the results presented in the paper. This is an important issue that needs to be addressed.

Reviewer #2 (Remarks to the Author):

This study uses a large ensemble of a low-resolution GCM to conclude that the observed snow trends in the western U.S. cannot be explained only with natural forcing. The GCM ensemble generally matches with the observed snow trends when all forcing are included. Further, the study projects a loss of up to 50% under the RCP8.5 scenario in the coming two decades, which has implications for snowmelt driven water resources in the western U.S.

Western U.S. historical snow trends and their projections under enhanced greenhouse gas forcing has been one of the most studied topic related with the North American climate. All the conclusions of this study about baseline trends and future accelerated decrease in cold season snow have already been well established through earlier studies (some of those are referenced in this study). Within this context, other than the use of a large initial condition ensemble of a GCM (or medium resolution ensemble of an RCM), there is nothing new or unique about this study that could warrant its suitability for journals like Nature Communications. Moreover, in my opinion, low-resolution GCM and medium resolution RCM ensemble are too coarse to provide any detail about the spatial heterogeneity of the observed trends or future responses to surface warming. Lumped analyses as those presented in this study are really not useful. In fact, comparison with SNOTEL in this way is not appropriate. In summary, while this study is good enough for publication in a more specific first tier journal, it is not novel enough for publication in Nature Communications.

Reviewer #3 (Remarks to the Author):

Review of 'Large near-term projected snowpack loss over the western United States'

Authors: J.C. Fyfe, C. Derksen, L. Mudryk, N.C. Swart, N.P. Molotch, G.M. Flato, X. Zhang, H. Wan, V.K. Arora, J. Scinocca and Y. Jiao

Year: 2016

Publication: Nature Communications

Review date: October 2016

Revision 0

Overview

Using point observations (SnoTel), four land surface models forced by meteorological observations (REANAL4) and two coupled atmosphere-ocean GCM ensembles for initial conditions (CanESM2; one ensemble with natural forcing only and one ensemble with both natural and anthropogenic forcing), the manuscript presents a quantification of past and future snowpack conditions in the western US for the periods 1982 – 2010 and 2008 – 2040. Collectively, the point observations show a regional decline in annual maximum SWE of ~10% (or -0.33cm/decade) from 1982 to 2016. The REANAL4 data shows a two-fold larger regional decline in modelled annual maximum SWE of ~22% for roughly the same time period (1982-2010). The CanESM2 climate model ensemble outputs show similar results to those obtained from the REANAL4 data, of ~16% regional decline in modelled annual maximum SWE for the period 1982 – 2010. Given that resulting the climate model trends generally agree with those from the observations-driven modelling (REANAL4), it is logical to then have confidence in future snowpack conditions as simulated by the climate model. A relatively wide spread of estimated future regional declines in modelled annual maximum SWE, as extracted from CanESM2 ensembles, are observed (2.5 – 47% reductions). However, the predicted decline in modelled annual maximum SWE from the ensemble mean of ~26% (2008 – 2040) is generally in accordance with the observed historic rates of decline and is an interesting and unique result. The authors then use the climate model ensembles (CanESM2) to confirm that the model can only replicate the observed snowpack declines when anthropogenic forcing is included. This is also an expected result and in general agreement with the published literature.

Overall, the study presents a logical modelling methodology with an impressive suite model realizations. The manuscript certainly presents a strong body of work to quantify the projected snowpack losses over the western US. After a quick literature review I found only two similar studies already in the literature that quantifies SWE decline to the period near future (Ashfaq et al., 2013, Casola et al., 2009). The manuscript provides a solid expansion of previous work and on this basis I am convinced that the main findings and contributions of this work will be of broad interest to the community. The paper could, however, be strengthened with enhanced comparison of results with previous work for the historical observational and land-modeling trend analysis. With relatively minor modifications I would recommend the manuscript to be sufficient quality for publication.

Minor comments

Why SnoTel and REANAL4 so different? I acknowledge that the authors are rightfully cautious when comparing point observations with gridded observations. However, the authors could reduce the spatial and temporal discrepancies for the comparison by comparing the datasets over the same time period (i.e. 1982-2010) and roughly at the same spatial sampling (i.e. the authors could pull out REANAL4 data that correspond only with SnoTel sites). The explanation of the reasons for the different scaling of the trends between the observations and the REANAL4 data could be stronger.

How were the initial conditions sampled? What parameters were changed for each of the 50 individual model realizations in the ensembles? An overview would suffice.

Why was the 2008 – 2040 trend in annual maximum SWE calculated differently from 1982 – 2010 evaluations? For the future trend calculation, the authors use a 'percent change' between two periods 2008 – 2018 and 2030 – 2040. Whereas, unless I am missing something here, for the historical (both observed and modelled trends) the authors use a 'linear trend' method for trend quantification.

At line 65 the authors cite reference #13 (Pierce et al., 2008) to support their findings of declining maximum annual SWE in the observations of -0.33 cm/decade. The referenced study is largely based on modelled SWE declines and detection-attribution of these trends rather than observation based trends. Perhaps the authors could compare their results to SWE trends from observations

such as those from the more recently published Harpold et al. (2012) which reports trends in annual maximum SWE ranging from -4.7 to +2.03 cm/decade (mean ~ -1.7 cm/decade). If appropriate, the authors should also explain why their results differ from values in other studies.

Similarly, the paper could be improved by comparing projected future trend values to those already in the published literature to leave the readers with an appreciation of 'where we stand' (for example, Casola et al., 2009 expected SWE declines by 2050; Brown and Mote, 2009 SWE declines by 2020, 2050 Figure 11 provided as suggestions only).

Consistent reporting of the Snotel observational period is required. At Line 374 the period reported is 1982-2014, but at Lines 58 and 62 the period of 1982-2016 is reported.

Line 150 – it is not clear how 'decadal variability' when combined with 'externally-forced declines' would in particular lead to large declines in snowpack. It is clear from the CanESM2 simulations that the external forcing is largely responsible for the recent declines (since the late 1980's). However, it is unclear how decadal variability, which is typically defined as >7-year variability that fluctuates both up and down with time, would further exacerbate snowpack decline in the long-term. Perhaps the authors are referring to instances when the phase of the atmospheric decadal variation is unfavourable for snowpack and combines with the external forcing for enhanced decline. I think this point is better characterized in the Concluding Remarks and the text in the main body of the manuscript could be improved to suit.

Line 160 – consider inserting a reference that describes the Aleutian Low in the context of North American climate for readers not familiar with general circulation features

Line 160-161 – 'the impact of warm air on regional snowpack is obvious', this may be true when air temperatures are above or about the freezing point, although a warming of 2°C in cold air temperatures i.e. from -10°C to -8°C may not really affect the snowpack as obviously (i.e. sublimation/melt). This sentence could be rephrased.

Lines 172-176 – this is a great section
Figures

Supplementary Figure 3 – the plot should have more geographical markers to allow the reader to immediately know which part of the globe is represented. The authors could do one or more of the following to improve clarity: use different line type/weight for land/ocean boundary and contour lines, insert lat/lon coordinate markers to the map, label in text certain land masses or ocean masses. This is one of the more interesting figures in the paper.

References

Ashfaq, M. et al. Near-term acceleration of hydroclimatic change in the western U.S. *J. Geophys. Res. Atmos.* 118, 10,676-10,693 (2013).

Casola, J. H. et al. Assessing the impacts of global warming on snowpack in the Washington cascades. *Journal of Climate* 22, 2758–2772 (2009).

Harpold, Adrian, et al. Changes in Snowpack Accumulation and Ablation in the Intermountain West. *Water Resources Research* 48 (11): W11501. (2012)

Pierce, D.W. et al. Attribution of declining western U.S. snowpack to human effects. *Journal of Climate* 21, 6425-6444 (2008).

We thank all of the reviewers for their very thoughtful comments. Reviewer comments are in plain text and our responses are in bold text.

Response to Reviewer #1

The paper addresses an interesting and timely topic and the paper is cleanly written and compact.

Thank you.

I only have two comments for the authors to consider (one is minor and one is major).

1. The authors present correlation statistics to indicate agreement between measured and modeled time series. Additional statistics that illustrate biases also are needed, or the authors can provide a justification for primarily focusing on variability rather than differences in magnitudes between measured and modeled time series.

We agree that additional statistics to illustrate biases are warranted. To this end, we have added two figures.

Figure 2: Compares SWE_{max} climatology on the horizontal axis. The average reanalysis value (pink circle) is less than the average observed value (green circle) which is less than the average simulated value (black circle). However, these values are not inconsistent with one another considering the uncertainties involved (pink bar, green bar and black ellipse, respectively). Figure 2 compares SWE_{max} trend on the vertical axis. Here, the average reanalysis loss is nearly identical to the average simulated loss; with both being about twice as large as the observed loss. However, as with climatology, all of the loss values are within uncertainty. These points are discussed in our revised manuscript in lines 114-126 and 142-144.

Figure 3: Compares the regional distribution of average reanalysis and average simulated SWE_{max} climatology. The simulated pattern is broadly consistent with that in the reanalyses, given the difference in spatial resolution and the smoothing resulting from ensemble averaging. This point is discussed in our revised manuscript in lines 144 to 147.

We thank Reviewer #1 for encouraging us to include this information.

2. The authors analyze annual maximum SWE. It would strengthen the paper if the authors also analyzed total winter snowfall. It could be that annual maximum SWE has decreased as shown in the study, but it also could be that more moderate SWE values have increased or become more prevalent for other months (e.g. the temporal distribution of snowfall and SWE during winter has become flatter). An examination of

total winter snowfall will confirm or refute the results presented in the paper. This is an important issue that needs to be addressed.

To address this good point we have added Supplementary Fig. 1. This figure shows that observed SWE has decreased in every month of the snow accumulation season and not just in April (the month typically associated with SWE_{max}). The reviewer suggests showing this with snowfall as well but since SWE is the hydrological variable connected to spring runoff, and precipitation is so poorly constrained in observations, we focus on SWE instead.

Response to Reviewer #2

This study uses a large ensemble of a low-resolution GCM to conclude that the observed snow trends in the western U.S. cannot be explained only with natural forcing. The GCM ensemble generally matches with the observed snow trends when all forcing are included. Further, the study projects a loss of up to 50% under the RCP8.5 scenario in the coming two decades, which has implications for snowmelt driven water resources in the western U.S.

Western U.S. historical snow trends and their projections under enhanced greenhouse gas forcing has been one of the most studied topic related with the North American climate. All the conclusions of this study about baseline trends and future accelerated decrease in cold season snow have already been well established through earlier studies (some of those are referenced in this study).

We disagree. Our study makes two important advances.

First, we detect and attribute an anthropogenic influence. A similar result was obtained in Pierce et al. (2008; our ref. 13) using earlier observations and models. Our study uses updated observations and models and is an important advance over Pierce et al. (2008).

Second, we quantify the roles of external forcing and internal variability. Using an unprecedented suite of global and regional climate model simulations we conclude that decadal variability added to externally forced response could result in a large loss of snowpack in the near future. This has not been established through earlier studies.

Finally, Reviewer #2 states "... future accelerated decrease in cold season snow have already been well established through earlier studies ...". To our knowledge, the only such study is Ashfaq et al. (2013; ref. 15 in our revised manuscript). In the Ashfaq et al. (2013) study "accelerated decrease" describes a situation where the average rate of loss (the forced response) over the near future is greater than the average rate of loss of over the near past. This is a very different situation

than the one considered in our study. The situation considered in our study is one where an episode of decadal variability in the near future, as obtained from a large ensemble of model simulations, adds to the average rate of change to produce a large loss. This is a much greater effect than the one described in Ashfaq et al. (2013). The difference in the magnitudes of these two effects can be seen visually in Fig. 5 of our revised manuscript. The Ashfaq et al. (2013) effect is seen as the small change in slope in the black curve between the first half of the analysis period and the second half of the analysis period. Our effect is seen as the large difference in slope between the red and black curves in the latter half of the analysis period.

In our revised manuscript in lines 202-212 we contrast the results of our study and the results of the Ashfaq et al. (2013) study. We note in particular the call that Ashfaq et al. (2013) make for large initial condition ensembles such as used in our study. We thank Reviewer #2 for pointing us in this direction and giving us the opportunity to further strengthen our manuscript.

Within this context, other than the use of a large initial condition ensemble of a GCM (or medium resolution ensemble of an RCM), there is nothing new or unique about this study that could warrant its suitability for journals like Nature Communications.

We disagree.

The use of our large ensembles of global and regional model simulations has allowed: 1) the detection of an anthropogenic influence in updated observations and reanalyses, and 2) the quantification of a potential contribution of decadal variability to snowpack loss in the near future. For the reasons outlined above, these are new and unique results that we feel are very suitable for reporting in Nature Communications.

Moreover, in my opinion, low-resolution GCM and medium resolution RCM ensemble are too coarse to provide any detail about the spatial heterogeneity of the observed trends or future responses to surface warming. Lumped analyses as those presented in this study are really not useful. In fact, comparison with SNOTEL in this way is not appropriate.

We disagree.

Our study is not concerned with spatial heterogeneity. We do not make local to area comparisons between Snotel measurements and reanalyses or climate model grid cells. Rather, our study is concerned with the coherent loss of snowpack observed across the entire region of the western United States. This coherency is seen in the widespread loss of observed snowpack shown in Fig. 1 of our revised manuscript. The fact that our global and regional climate models reproduce the region wide loss within uncertainty (as shown in Fig. 2 of our

revised manuscript) is strong evidence that these models are appropriate tools for such an investigation.

In summary, while this study is good enough for publication in a more specific first tier journal, it is not novel enough for publication in Nature Communications.

For the reasons stated above, we disagree that our study is not novel enough for publication in Nature Communications, and we have made revisions (in response to other review comments) that we believe further strengthen the manuscript.

Response to Reviewer #3

Overview

Using point observations (SnoTel), four land surface models forced by meteorological observations (REANAL4) and two coupled atmosphere-ocean GCM ensembles for initial conditions (CanESM2; one ensemble with natural forcing only and one ensemble with both natural and anthropogenic forcing), the manuscript presents a quantification of past and future snowpack conditions in the western US for the periods 1982 – 2010 and 2008 – 2040. Collectively, the point observations show a regional decline in annual maximum SWE of ~10% (or -0.33cm/decade) from 1982 to 2016. The REANAL4 data shows a two-fold larger regional decline in modelled annual maximum SWE of ~22% for roughly the same time period (1982-2010). The CanESM2 climate model ensemble outputs show similar results to those obtained from the REANAL4 data, of ~16% regional decline in modelled annual maximum SWE for the period 1982 – 2010. Given that resulting the climate model trends generally agree with those from the observations-driven modelling (REANAL4), it is logical to then have confidence in future snowpack conditions as simulated by the climate model. A relatively wide spread of estimated future regional declines in modelled annual maximum SWE, as extracted from CanESM2 ensembles, are observed (2.5 – 47% reductions). However, the predicted decline in modelled annual maximum SWE from the ensemble mean of ~26% (2008 – 2040) is generally in accordance with the observed historic rates of decline and is an interesting and unique result. The authors then use the climate model ensembles (CanESM2) to confirm that the model can only replicate the observed snowpack declines when anthropogenic forcing is included. This is also an expected result and in general agreement with the published literature.

Overall, the study presents a logical modelling methodology with an impressive suite model realizations. The manuscript certainly presents a strong body of work to quantify the projected snowpack losses over the western US.

Thank you.

After a quick literature review I found only two similar studies already in the literature that quantifies SWE decline to the period near future (Ashfaq et al., 2013, Casola et al., 2009).

Thank you for bringing these studies to our attention. We discuss and/or refer to these studies in lines 210-212 and 221-223 of our revise manuscript. Please also see our response to Reviewer #2 regarding the Ashfaq et al. (2013) study and our comments below regarding the Casola et al. (2009) study.

The manuscript provides a solid expansion of previous work and on this basis I am convinced that the main findings and contributions of this work will be of broad interest to the community.

Thank you.

The paper could, however, be strengthened with enhanced comparison of results with previous work for the historical observational and land-modeling trend analysis. With relatively minor modifications I would recommend the manuscript to be sufficient quality for publication.

We have enhanced our comparison with other studies of past and future change (see details below). This has strengthened the context of our results.

Minor comments

Why SnoTel and REANAL4 so different? I acknowledge that the authors are rightfully cautious when comparing point observations with gridded observations. However, the authors could reduce the spatial and temporal discrepancies for the comparison by comparing the datasets over the same time period (i.e. 1982-2010) and roughly at the same spatial sampling (i.e. the authors could pull out REANAL4 data that correspond only with SnoTel sites). The explanation of the reasons for the different scaling of the trends between the observations and the REANAL4 data could be stronger.

We were remiss in not addressing this difference. Figure 2 of our revised manuscript compares the ranges of estimates of SWE_{max} climatology and trend. The green bars are uncertainties as computed as \pm one standard deviation across 10,000 random samples of the SnoTel stations. In each case, a station is randomly selected and the average climatology and trend is calculated across a random selection of no more than 10 of its neighboring stations within 1° longitude and 1° latitude (the spatial resolution of the reanalyses). The pink bars represent structural uncertainty in the reanalyses. From this figure we conclude that the observed and reanalysis climatologies and trends are not inconsistent. That is to say, the ranges of observed and reanalysis values are overlapping. This is discussed in our revised manuscript in lines 113 to 126.

How were the initial conditions sampled? What parameters were changed for each of the 50 individual model realizations in the ensembles? An overview would suffice.

In an aspect of the cloud parametrization in the model there is a random number generator that uses a preset seed. The 50 individual model realizations are based on different seeds. In this way, different realizations were produced without affecting the structure of the model. An overview of our large ensemble of simulations is presented in the Methods section of our revised manuscript.

Why was the 2008 – 2040 trend in annual maximum SWE calculated differently from 1982 – 2010 evaluations? For the future trend calculation, the authors use a ‘percent change’ between two periods 2008 – 2018 and 2030 – 2040. Whereas, unless I am missing something here, for the historical (both observed and modelled trends) the authors use a ‘linear trend’ method for trend quantification.

In our revised manuscript we present historical and future results as linear trends and as percent differences. Percent differences are defined as late period minus early period divided by early period.

At line 65 the authors cite reference #13 (Pierce et al., 2008) to support their findings of declining maximum annual SWE in the observations of -0.33 cm/decade. The referenced study is largely based on modelled SWE declines and detection-attribution of these trends rather than observation based trends. Perhaps the authors could compare their results to SWE trends from observations such as those from the more recently published Harpold et al. (2012) which reports trends in annual maximum SWE ranging from -4.7 to +2.03 cm/decade (mean ~ -1.7 cm/decade). If appropriate, the authors should also explain why their results differ from values in other studies.

The results in the Harpold et al. (2012; ref. 16 in our revised manuscript) study show that Snotel trend variability can be high when computed for individual stations, but are more consistent when sites are averaged for the type of regional analysis we employ. Harpold et al. (2012) also show that differences in trend results can also be due to the specific SWE related variable that is analyzed (e.g., using a fixed date such as 1 April, or daily or monthly SWE_{max}). This is discussed in lines 90-91 of our revised manuscript.

Similarly, the paper could be improved by comparing projected future trend values to those already in the published literature to leave the readers with an appreciation of ‘where we stand’ (for example, Casola et al., 2009 expected SWE declines by 2050; Brown and Mote, 2009 SWE declines by 2020, 2050 Figure 11 provided as suggestions only).

Note that the Casola et al. (2009; ref. 27 in our revised manuscript) study focuses only on the Cascades and the Brown and Mote (2009; ref. 28 in our revised manuscript) study shows only model consensus on trend sign, so we cannot

directly compare the results of our study to this earlier work. Still, in lines 222-223 we use their results as illustrations of the sensitivity of projected SWE changes to the range of simulated temperature and precipitation.

Consistent reporting of the SnoTel observational period is required. At Line 374 the period reported is 1982-2014, but at Lines 58 and 62 the period of 1982-2016 is reported.

This has been corrected in our revised manuscript.

Line 150 – it is not clear how ‘decadal variability’ when combined with ‘externally-forced declines’ would in particular lead to large declines in snowpack. It is clear from the CanESM2 simulations that the external forcing is largely responsible for the recent declines (since the late 1980’s). However, it is unclear how decadal variability, which is typically defined as >7-year variability that fluctuates both up and down with time, would further exacerbate snowpack decline in the long-term. Perhaps the authors are referring to instances when the phase of the atmospheric decadal variation is unfavorable for snowpack and combines with the external forcing for enhanced decline. I think this point is better characterized in the Concluding Remarks and the text in the main body of the manuscript could be improved to suit.

Thank you for drawing this to our attention. This has been clarified in our revised manuscript.

Line 160 – consider inserting a reference that describes the Aleutian Low in the context of North American climate for readers not familiar with general circulation features.

This is good suggestion that we have adopted in lines 257-260 of our revised manuscript.

Line 160-161 – ‘the impact of warm air on regional snowpack is obvious’, this may be true when air temperatures are above or about the freezing point, although a warming of 2°C in cold air temperatures i.e. from -10°C to -8°C may not really affect the snowpack as obviously (i.e. sublimation/melt). This sentence could be rephrased.

This sentence has been rephrased in lines 263-264 in our revised manuscript.

Lines 172-176 – this is a great section

Thank you.

Figures

Supplementary Figure 3 – the plot should have more geographical markers to allow the reader to immediately know which part of the globe is represented. The authors could

do one or more of the following to improve clarity: use different line type/weight for land/ocean boundary and contour lines, insert lat/lon coordinate markers to the map, label in text certain land masses or ocean masses. This is one of the more interesting figures in the paper.

Thank you.

We agree that this is an interesting figure. We have improved this figure and have moved it into the main manuscript of our revised manuscript (as Figure 6).

REVIEWERS' COMMENTS:

Reviewer #1 (Remarks to the Author):

The authors have adequately addressed all of my previous concerns and the paper is acceptable for publication.

Reviewer #2 (Remarks to the Author):

This is a manuscript that reviewed previously. I have now gone through the changes that authors have made in response to each of the reviewers' comments and while I still have reservations, I am ok with the revisions that authors have made in the current version and in general satisfied to a level that I recommend it for publication without any further changes.

Reviewer #3 (Remarks to the Author):

The authors were graciously responsive to the suggestions that I made in my initial review in October 2016. I am generally satisfied that my initial points of concern have now been addressed in the revised manuscript.

I have some minor edits/suggestions/comments upon reading the revised manuscript:

- Did you notice any systematic biases one way or another in simulated SWE between the four models in the REANAL4 group? If there is something strong there it might be worth mentioning near lines 107-108.
- Regarding Figure 2, I would expect the climatological SWE from the REANAL4 to be lower than that observed in the SNOTEL suite due to resolved elevation (smoothing and lowering) in the models. However, it is nice to see that the two climatology estimates are within error estimates (as pointed out by the authors at Line 114). The interesting thing about this figure is that it clearly shows that most of the CanESM2 climatological estimates are much higher than REANAL4 and SNOTEL. I'm not sure I saw an explanation of this in the text.
- The x axis labels in Figure 4 appear to have been trimmed
- I would still add Lat/Lon coordinates to Figure 6.

Reviewer comments are in plain text and our responses are in bold text.

Response to Reviewer #3

Did you notice any systematic biases one way or another in simulated SWE between the four models in the REANAL4 group? If there is something strong there it might be worth mentioning near lines 107-108.

**We have added the following sentence in our revised manuscript:
“Differences between these datasets have been discussed at length in ref. 17.”**

Regarding Figure 2, I would expect the climatological SWE from the REANAL4 to be lower than that observed in the SNOTEL suite due to resolved elevation (smoothing and lowering) in the models. However, it is nice to see that the two climatology estimates are within error estimates (as pointed out by the authors at Line 114). The interesting thing about this figure is that it clearly shows that most of the CanESM2 climatological estimates are much higher than REANAL4 and SNOTEL. I'm not sure I saw an explanation of this in the text.

We have added the following sentence in our revised manuscript: “The latter suggests greater climatological snowfall in the model than in reality, although this is difficult to assess given the absence of a reliable long-term observational record of winter precipitation.”

The x axis labels in Figure 4 appear to have been trimmed.

The x axis labels are intact.

I would still add Lat/Lon coordinates to Figure 6.

Done.